# Inguinal Fat Compensates Whole Body Metabolic Functionality in Partially Lipodystrophic Mice with Reduced PPARγ Expression

**DOI:** 10.3390/ijms24043904

**Published:** 2023-02-15

**Authors:** Cherng-Shyang Chang, Shang-Shiuan Yu, Li-Chun Ho, Shu-Hsin Chao, Ting-Yu Chou, Ai-Ning Shao, Ling-Zhen Kao, Chia-Yu Chang, Yu-Hsin Chen, Ming-Shan Wu, Pei-Jane Tsai, Nobuyo Maeda, Yau-Sheng Tsai

**Affiliations:** 1Institute of Clinical Medicine, College of Medicine, National Cheng Kung University, Tainan 701, Taiwan; 2Institute of Basic Medical Sciences, College of Medicine, National Cheng Kung University, Tainan 701, Taiwan; 3School of Medicine, College of Medicine, I-Shou University, Kaohsiung 824, Taiwan; 4Division of General Medicine, Department of Internal Medicine, E-DA Hospital, Kaohsiung 824, Taiwan; 5Department of Physiology, College of Medicine, National Cheng Kung University, Tainan 701, Taiwan; 6Institute of Molecular Medicine, College of Medicine, National Cheng Kung University, Tainan 701, Taiwan; 7Department of Medical Laboratory Science and Biotechnology, College of Medicine, National Cheng Kung University, Tainan 701, Taiwan; 8Department of Pathology and Laboratory Medicine, University of North Carolina, Chapel Hill, NC 27599, USA; 9Clinical Medicine Research Center, National Cheng Kung University Hospital, Tainan 704, Taiwan

**Keywords:** PPARγ, partial lipodystrophy, insulin sensitivity, fat depot, adipose tissue flexibility, inguinal fat-dependence index

## Abstract

Peroxisome proliferator-activated receptor γ (PPARγ) gene mutations in humans and mice lead to whole-body insulin resistance and partial lipodystrophy. It is unclear whether preserved fat depots in partial lipodystrophy are beneficial for whole-body metabolic homeostasis. We analyzed the insulin response and expression of metabolic genes in the preserved fat depots of *Pparg^C/-^* mice, a familial partial lipodystrophy type 3 (FPLD3) mouse model resulting from a 75% decrease in *Pparg* transcripts. Perigonadal fat of *Pparg^C/-^* mice in the basal state showed dramatic decreases in adipose tissue mass and insulin sensitivity, whereas inguinal fat showed compensatory increases. Preservation of inguinal fat metabolic ability and flexibility was reflected by the normal expression of metabolic genes in the basal or fasting/refeeding states. The high nutrient load further increased insulin sensitivity in inguinal fat, but the expression of metabolic genes became dysregulated. Inguinal fat removal resulted in further impairment of whole-body insulin sensitivity in *Pparg^C/-^* mice. Conversely, the compensatory increase in insulin sensitivity of the inguinal fat in *Pparg^C/-^* mice diminished as activation of PPARγ by its agonists restored insulin sensitivity and metabolic ability of perigonadal fat. Together, we demonstrated that inguinal fat of *Pparg^C/-^* mice plays a compensatory role in combating perigonadal fat abnormalities.

## 1. Introduction

Lipodystrophy is a heterogeneous group of rare genetic disorders characterized by varying degrees of fat loss and metabolic abnormalities [1]. Fat loss is classified as generalized (involving all body fat depots), partial (affecting the limbs), or localized (from discrete areas of the body) [2]. Based on the pattern of adipose tissue loss, lipodystrophy can be categorized into either congenital generalized lipodystrophy, a general lack of adipose tissue present at birth or becoming apparent in early childhood, or familial partial lipodystrophy (FPLD), normal adipose tissue distribution at birth and throughout childhood but lacking adipose tissue in the arms and legs during adolescence [3]. FPLD type 3 (FPLD3, OMIM #604367) is dominantly inherited and is caused by mutations in the *PPARG* gene [4,5,6]. Peroxisome proliferator-activated receptor γ, PPARγ, is a nuclear receptor highly expressed in adipose tissue and is critical for mediating lipogenesis and adipogenesis. While homozygous *Pparg* knockout mice (*Pparg^-/-^*) are embryonically lethal [7,8], adipocyte-specific knockout mice survive and show generalized decreases in all body fat depots [9,10]. In contrast, previous studies have shown that FPLD3 patients with various PPARG mutations frequently experience loss of subcutaneous fat from the limbs and gluteal regions, but relatively preserved subcutaneous and visceral abdominal fat depots [11,12]. We previously established that *Pparg^C/-^* mice, with a 75% reduction in *Pparg* transcript levels, exhibited partial lipodystrophy with loss of perigonadal fat of the visceral depot, and normal or compensatory hypertrophy of inguinal fat of the subcutaneous depot [13,14]. While loss of fat in FPLD3 is evidenced in both human and mouse models, it is still unclear why the pattern of fat loss is regional and why some depots are spared and not others [15]. Thus, it is critical to identify the underlying mechanisms by which PPARγ alters adipose tissue homeostasis in partial lipodystrophy.

The normal function of adipose tissue is to buffer the daily influx of dietary fat into circulation [16]. In the postprandial period, adipose tissue is known to respond to plasma insulin to alter its metabolic status, including a decrease in lipolysis and increases in lipid uptake, lipogenesis, and de novo lipogenesis, and excess energy is stored in the form of triglycerides [17]. In the fasting state, adipose tissue releases fatty acids into circulation by increasing lipolysis to supply energy to other tissues. The input and output of lipids in adipose tissue reflect the flexibility of adipose tissue in controlling energy balance. Adipose tissue flexibility is also affected by adipogenesis, in which increased adipocytes provide additional capacity for lipid storage. However, if the maximum capacity of adipose is reached, the ectopic lipid accumulation occurs in tissues such as liver and skeletal muscle, and exacerbates the development of insulin resistance [18,19]. In addition, mice with lipolysis deficiency also showed decreased storage capacity in adipose tissue, increased lipid accumulation in the liver, and increased plasma insulin levels [20]. Thus, adipose tissue flexibility is an important factor for maintaining metabolic homeostasis. Interestingly, mice associated with FPLD3 showed a loss of perigonadal fat and insulin resistance [13,21,22], suggesting that the loss of adipose tissue capacity from perigonadal fat is linked to an imbalance in whole-body metabolism. Because inguinal fat is preserved in mice with FPLD3, it is intriguing to postulate that the adipose tissue capacity of inguinal fat is retained, or indeed this depot is dispensable for this metabolic imbalance.

A mouse model with transplantation of inguinal fat demonstrated that subcutaneous adipose tissue could have direct and beneficial effects on the control of metabolism [23]. It has been shown that insulin-stimulated antilipolytic effects remained in inguinal fat under high-fat diet (HFD) treatment, whereas this effect was lost in perigonadal fat [24]. Our previous study showed that inguinal fat in mice is less dependent on PPARγ-mediated signaling than perigonadal fat [13]. PPARγ agonists redistribute fat from visceral to subcutaneous depots by stimulating lipid uptake and esterification potential in the inguinal fat [25]. These results suggest that inguinal fat has higher flexibility than perigonadal fat in response to a high nutrient load, low mediator conditions, and stimulator activation. Thus, it is reasonable to speculate that inguinal fat in FPLD3 undertakes insulin-mediated actions to maintain metabolic homeostasis.

To test the hypothesis that the inguinal fat in FPLD3 plays an indispensable role in maintaining whole-body metabolic homeostasis, we analyzed the insulin response and expression of metabolic genes in inguinal fat of *Pparg^C/-^* mice, an FPLD3 mouse model caused by a 75% decrease in the *Pparg* gene expression. Inguinal fat lipectomy was performed to test the contribution of inguinal fat in mice with partial lipodystrophy. We also examined the responses of these fat depots to the PPARγ agonist thiazolidinediones (TZD) to establish an association between PPARγ activation and inguinal fat dependence. Our findings show a compensatory role for inguinal fat in the FPLD3 mouse model, in which inguinal fat showed enhanced insulin signaling and its contribution to whole-body insulin sensitivity.

## 2. Results

### 2.1. Compensatory Increase in Insulin Sensitivity in Inguinal Fat of Pparg^C/-^ Mice with Partial Lipodystrophy

Male *Pparg^C/-^* mice exhibited partial lipodystrophy, which was reflected by a dramatic decrease in perigonadal fat weight (Figure 1A,B). The weights of other visceral depots such as the mesentery and retroperitoneal fat in *Pparg^C/-^* mice were not altered. However, subcutaneous fat depots such as inguinal and scapular fat showed compensatory increases in fat weight in *Pparg^C/-^* mice compared to wild-type mice, suggesting that lipid storage was increased in subcutaneous fat in response to the loss of storage capacity of perigonadal fat.

Another characteristic of *Pparg^C/-^* mice is whole-body insulin resistance. While glucose levels in *Pparg^C/-^* mice did not change during the glucose tolerance test (GTT) compared with those of wild-type mice (Figure 1C), plasma insulin levels of *Pparg^C/-^* mice were increased (Figure 1D), leading to a significant increase in the insulin resistance (IR index in *Pparg^C/-^* mice (Figure 1E). This decrease in whole-body insulin sensitivity of *Pparg^C/-^* mice was associated with a decrease in insulin sensitivity of perigonadal fat, as reflected by the reduction in Akt phosphorylation of perigonadal fat in *Pparg^C/-^* mice after insulin stimulation (Figure 1F). In contrast, insulin-induced Akt phosphorylation was increased in the inguinal fat of *Pparg^C/-^* mice compared with that in wild-type mice (Figure 1G), suggesting a compensatory effect of inguinal fat in *Pparg^C/-^* mice. Other insulin-responsive tissues, such as the liver and skeletal muscle, showed no change in insulin-induced Akt phosphorylation in *Pparg^C/-^* mice compared with wild-type mice (Figure 1H,I). Together, these results indicate that whole-body insulin resistance in *Pparg^C/-^* mice is primarily due to the loss of storage capacity and insulin sensitivity of perigonadal fat. Inguinal fat plays a compensatory role against abnormalities in perigonadal fat in *Pparg^C/-^* mice.

### 2.2. Preserved Metabolic Ability of Inguinal Fat in Pparg^C/-^ Mice

To test whether metabolic ability was altered in different fat depots of mice with partial lipodystrophy, we next examined the mRNA levels of metabolic genes in *Pparg^C/-^* mice (Figure 1J and Appendix A). Because C/EBPδ is an upstream factor of PPARγ in the transcriptional cascade during adipogenesis, increased C/EBPδ expression in response to PPARγ mutation was expected. This was the case for perigonadal fat; however, the increase in C/EBPδ transcripts was not as significant in inguinal fat. Similarly, PGC1α was significantly increased in perigonadal fat, but not in inguinal fat. Instead, the reduced *Pparg* expression decreased other genes involved in adipogenesis (SREBP1, CEBPA, and CEBPB) in the perigonadal fat of *Pparg^C/-^* mice. The mRNA levels of genes associated with lipid uptake (LPL, AQP7, and CD36) were also decreased in the perigonadal fat of *Pparg^C/-^* mice, accompanied by a decrease in genes associated with lipogenesis (FABP4, ACSL1, GPAT1, and DGAT1) and de novo lipogenesis (GLUT4, PEPCK, ACC1, FAS, and SCD1). However, the relatively lesser extents of decrease were found in the mRNA levels of genes associated with adipogenesis, lipid uptake, lipogenesis, and de novo lipogenesis in the inguinal fat of *Pparg^C/-^* mice compared to those in perigonadal fat. The expression of LPL is significantly increased in the inguinal fat of *Pparg^C/-^* mice. Collectively, these results suggest that the adipose tissue capacity of inguinal fat is higher than that of perigonadal fat in mice with partial lipodystrophy. Similarly, the mRNA levels of genes involved in lipolysis (HSL and ATGL) were largely decreased in perigonadal fat in *Pparg^C/-^* mice and relatively normal in inguinal fat. The genes associated with insulin signaling (IRS1 and IRS2) and adipokines (adiponectin and leptin) in adipose tissue were downregulated in perigonadal fat of *Pparg^C/-^* mice but were relatively normal in inguinal fat. Notably, the mRNA level of gene associated with adipose fibrosis (COL1A1) was largely increased in the perigonadal fat of *Pparg^C/-^* mice, but this effect was not observed in the inguinal fat depot of *Pparg^C/-^* mice. Together, the inguinal fat of *Pparg^C/-^* mice showed higher levels of genes associated with adipose metabolism and lower levels of genes associated with adipose fibrosis relative to perigonadal fat, suggesting preserved metabolic ability of inguinal fat in mice with partial lipodystrophy.

### 2.3. Preserved Adipose Tissue Flexibility of Inguinal Fat in Pparg^C/-^ Mice

Because the metabolic ability of fat can be linked to adipose tissue flexibility, we examined whether adipose tissue flexibility of individual fat was altered in *Pparg^C/-^* mice. In both perigonadal fat and inguinal fat of wild-type mice, lipid mobilization was activated after 24 h of fasting, reflected by increases in the mRNA levels of HSL, CD36, and LPL, and then inhibited after 2 h of refeeding (Figure 2A). We found that lipid mobilization was largely decreased in the perigonadal fat of *Pparg^C/-^* mice. In contrast, lipid mobilization of inguinal fat in *Pparg^C/-^* mice was comparable to that in wild-type mice. The ex vivo lipolysis analysis also showed a decrease in fatty acid secretion in the perigonadal fat of *Pparg^C/-^* mice, but this decrease was not observed in inguinal fat (Figure 2B). Together, these results suggest that adipose tissue flexibility is impaired in perigonadal fat, but remains intact in inguinal fat of *Pparg^C/-^* mice. Using Masson’s trichrome or picrosirius red staining, we found perigonadal fat of *Pparg^C/-^* mice exhibited more positive signals than wild-type mice, indications of increased fibrosis in perigonadal fat of *Pparg*^C/-^ mice (Appendix A). In contrast, the similar extent of fibrosis staining was observed between the inguinal fat of *Pparg*^C/-^ and wild-type mice, suggesting the inguinal fat of *Pparg*^C/-^ mice have normal extracellular structure.

### 2.4. High-Fat Diet Enhances Insulin Sensitivity in Inguinal Fat of Pparg^C/-^ Mice

To test the effect of increased nutrient load on inguinal fat in partial lipodystrophy, *Pparg^C/-^* mice were fed a high-fat diet (HFD), and the insulin-induced signal was examined. HFD-fed *Pparg^C/-^* mice showed increased plasma glucose and insulin levels during the glucose tolerance test (Figure 3A,B), leading to an increase in the IR index (Figure 3C). The perigonadal fat of HFD-fed *Pparg^C/-^* mice showed a 2.5-fold decrease in insulin-induced Akt phosphorylation compared to that in wild-type mice (Figure 3D). In contrast, a 1.8-fold increase in insulin-induced Akt phosphorylation was observed in the inguinal fat of HFD-fed *Pparg^C/-^* mice (Figure 3E). In comparison with *Pparg^C/-^* mice fed with regular chow, the imbalance of Akt phosphorylation was largely increased in the fat of HFD-fed *Pparg^C/-^* mice. In addition, insulin-induced Akt phosphorylation was slightly decreased in the liver and skeletal muscle of HFD-fed *Pparg^C/-^* mice (Figure 3F,G), suggesting that the liver and skeletal muscle were affected by a high nutrient load in mice with partial lipodystrophy. These results indicate that HFD exacerbates the imbalance in insulin sensitivity in the fat, liver, and skeletal muscle of mice with partial lipodystrophy. The compensatory increase in insulin sensitivity in inguinal fat is associated with an increased nutrient load.

### 2.5. High-Fat Diet Exacerbates Abnormality of Metabolic Ability in Inguinal Fat of Pparg^C/-^ Mice

To study whether increased nutrient load affects the metabolic ability of inguinal fat, the mRNA levels of metabolic genes in HFD-fed *Pparg^C/-^* mice were examined. The mRNA levels of genes involved in adipogenesis (SREBP1, CEBPA, and CEBPB), lipid uptake (LPL, AQP7, and CD36), lipogenesis (FABP4, ACSL1, GPAT1, and DGAT1), de novo lipogenesis (GLUT4, PEPCK, ACC1, FAS, and SCD1), and lipolysis (HSL and ATGL) were downregulated in the perigonadal fat of HFD-fed *Pparg^C/-^* mice (Figure 3H and Appendix A). Notably, these genes were also downregulated in inguinal fat to the same extent in perigonadal fat of HFD-fed *Pparg^C/-^* mice, suggesting that the metabolic ability of inguinal fat was also compromised by high nutrient load in mice with partial lipodystrophy. While the mRNA level of leptin was not altered by HFD, the expression of adiponectin was decreased in the inguinal fat of HFD-fed *Pparg^C/-^* mice. In addition, the mRNA levels of metabolic genes in the liver and skeletal muscle were modestly altered (Figure 3I,J and Appendix A). Together, these results show that a high nutrient load dysregulates metabolic genes and impairs the metabolic ability of inguinal fat in mice with partial lipodystrophy.

### 2.6. Inguinal Fat Contributes to the Whole-Body Insulin Sensitivity in Pparg^C/-^ Mice

The increased insulin sensitivity and decreased metabolic ability of inguinal fat in HFD-fed *Pparg^C/-^* mice prompted us to test the contribution of inguinal fat to maintaining whole-body insulin sensitivity. To evaluate the dependency of inguinal fat on whole-body insulin sensitivity, surgical removal of the inguinal fat was performed. No obvious differences in glucose and insulin levels and IR index were found between inguinal fat-removed and sham-operated wild-type mice after 3 weeks of HFD treatment (Figure 4A–C). In contrast, inguinal fat-removed *Pparg^C/-^* mice showed evident increases in glucose and insulin levels and IR index compared with sham-operated *Pparg^C/-^* mice after 3 weeks of HFD treatment. These results suggest that *Pparg^C/-^* mice have a higher dependency on inguinal fat than wild-type mice for maintaining whole-body insulin sensitivity.

### 2.7. TZD Treatment Reverses Insulin Sensitivity and Metabolic Ability of Fat Tissue in Pparg^C/-^ Mice

To test whether TZD treatment improves insulin sensitivity in mice with partial lipodystrophy, *Pparg^C/-^* mice and wild-type mice were fed an HFD supplemented with rosiglitazone (HFDTZD) for 4 weeks. The plasma glucose levels of HFDTZD-fed *Pparg^C/-^* mice were higher than those of HFDTZD-fed wild-type mice during GTT, whereas the plasma insulin levels of HFDTZD-fed *Pparg^C/-^* and wild-type mice were similar (Figure 5A,B). The whole-body insulin sensitivity of HFDTZD-fed *Pparg^C/-^* mice was reversed to the normal level, as reflected by a similar IR index between HFDTZD-fed *Pparg^C/-^* and wild-type mice (Figure 5C). HFDTZD-fed *Pparg^C/-^* mice showed similar insulin-induced Akt phosphorylation in both perigonadal and inguinal fat (Figure 5D,E), suggesting that TZD is effective in correcting insulin sensitivity of fat depots in mice with partial lipodystrophy. The liver and skeletal muscle of HFDTZD-fed *Pparg^C/-^* mice also showed insulin-induced Akt phosphorylation similar to that of wild-type mice (Figure 5F,G), suggesting that TZD is effective in normalizing the insulin sensitivity of the liver and skeletal muscle in mice with partial lipodystrophy.

Interestingly, TZD treatment corrected most of the downregulated genes in both the perigonadal and inguinal fat of HFD-fed *Pparg^C/-^* mice (Figure 5H and Appendix A). Moreover, TZD treatment elicited a greater upregulation of genes associated with adipogenesis, lipid uptake, and lipogenesis in perigonadal fat than in inguinal fat of *Pparg^C/-^* mice. These results suggest that the residual perigonadal fat of *Pparg^C/-^* mice, although being attenuated in capacity and function, remains sensitive, and even more potent, to TZD treatment. Although less improvement in metabolic gene expression by TZD treatment was found in the liver of *Pparg^C/-^* mice, they exhibited compensatory increases by TZD in skeletal muscle (Figure 5I,J and Appendix A). Together, these results suggest that TZD is effective in correcting the metabolic ability in both residual perigonadal fat and preserved inguinal fat.

### 2.8. Inguinal Fat-Dependence Index Reflects Adipose Tissue Homeostasis

To evaluate the response of TZD to different depots in mice with lipodystrophy, we compared insulin-induced Akt phosphorylation in perigonadal and inguinal fat with or without TZD treatment. In wild-type mice, insulin sensitivity was largely increased by TZD treatment in inguinal fat (2.15×), but this increase was not found in perigonadal fat (1.05×, Figure 6). Thus, wild-type mice showed an increase in inguinal fat dependency, reflected by a 2.04-fold (2.15/1.05 = 2.04) increase in Akt phosphorylation in inguinal fat (2.15×) compared with perigonadal fat (1.05×). This increase in inguinal fat dependency can be linked to the beneficial effects of TZD on the regulation of metabolic ability. The loss of perigonadal fat in HFD-fed *Pparg^C/-^* mice induced a larger increase in inguinal fat dependency, as reflected by the 4.55-fold increase in Akt phosphorylation in inguinal fat (2.18×) divided by perigonadal fat (0.47×). This abnormality of adipose tissue homeostasis in *Pparg^C/-^* mice was effectively reversed by TZD treatment, in which the inguinal fat-dependence index was decreased from 4.55 to 2.78 and closed to the index 2.04 of wild-type mice. Thus, the inguinal fat-dependence index clearly reflects adipose tissue homeostasis. A modest increase in the inguinal fat-dependence index indicated the competent function of inguinal fat in balancing adipose homeostasis. However, a large increase in the inguinal fat-dependence index indicated overloading of inguinal fat and an imbalance of adipose tissue homeostasis. Together, the workload of inguinal fat, defined as the inguinal fat-dependence index, is a critical determinant of adipose tissue homeostasis, and TZD is effective in reducing the inguinal fat workload in partial lipodystrophy.

## 3. Discussion

Our results showed that inguinal fat plays a compensatory role in partial lipodystrophy by compensating for the loss of perigonadal fat. We found that inguinal fat in *Pparg^C/-^* mice with partial lipodystrophy exhibited increased insulin sensitivity and preserved adipose tissue flexibility, despite significant decreases in these in perigonadal fat. Moreover, the high nutrient load further increased the insulin sensitivity of inguinal fat in *Pparg^C/-^* mice, although the increased workload ultimately dampened the metabolic ability of inguinal fat to the same extent as perigonadal fat. The importance of the increased insulin sensitivity of the inguinal fat was also evidenced by its removal which resulted in impairment of whole-body insulin sensitivity in *Pparg^C/-^* mice. The activation of PPARγ by TZD released an imbalance in insulin sensitivity between perigonadal and inguinal fat, and metabolic ability was restored in these depots. We further demonstrated that TZD increased the inguinal dependency in wild-type mice. The increased workload of inguinal fat under HFD feeding was confirmed by a large increase in inguinal dependency in *Pparg^C/-^* mice; however, this abnormal increase in inguinal dependency was attenuated by TZD treatment. These results underscore that PPARγ is critical for managing adipose tissue homeostasis, balancing the workload between fat depots.

The removal of perigonadal fat from mice improved insulin sensitivity [26,27,28]. However, the loss of perigonadal fat in mice with partial lipodystrophy impaired insulin sensitivity [13,21,22]. This controversy suggested that the metabolic condition of tissues other than the perigonadal depot was critical for whole-body insulin sensitivity in partial lipodystrophy. Moreover, increased subcutaneous fat was associated with the ability to absorb excess lipids, avoiding the accumulation of visceral fat that drives insulin resistance [29]. In the present study, we found that inguinal fat in partial lipodystrophy showed a slight decrease in metabolic ability, which was exacerbated by an HFD. By testing local insulin sensitivity, we found a compensatory role for inguinal fat under normal nutrient workload, and its compensatory effect was exaggerated after increased nutrient workload. These results suggest a predominant role for inguinal fat in the maintenance of whole-body insulin sensitivity in patients with partial lipodystrophy. Subcutaneous fat or inguinal fat is preserved in partial lipodystrophy in humans and mice, suggesting that the adipose tissue capacity of these depots is competent in partial lipodystrophy. Thus, the improvement of adipose function in residual fat is a strategy for maintaining insulin sensitivity in patients with partial lipodystrophy.

For the compensatory roles of other visceral depots in combating perigonadal fat abnormalities, we examined the dynamics of fat weights during the 24 h of fasting and 2 h of refeeding course (Appendix A). The lower weights of perigonadal fat and higher weights of inguinal fat stay consistent throughout the fasting/refeeding course. However, the mesentery fat weight of *Pparg^C/-^* mice exhibits a steeper drop than that of wild-type mice. The difference of retroperitoneal fat weights between genotypes was not consistent throughout the fasting/refeeding course. In addition, the fat weight of perigonadal was about 2~3 times of other visceral fat depots. Because of the consistency of fat weight during the fasting/refeeding course and the relatively higher mass of perigonadal fat, we did not examine the compensatory roles of other visceral fat depots. However, it cannot be excluded for this possibility.

Several groups have shown that PPARγ activation can influence insulin signaling at various steps, resulting in improved whole-body insulin sensitivity and enhanced glucose and lipid metabolism. While these results demonstrated the effect of PPARγ activation on insulin sensitivity in adipocytes, the difference between fat depots is less clear. In the present study, we found that TZD activated insulin-induced Akt phosphorylation in the inguinal fat (2.15×, Figure 6), but not in the perigonadal fat (1.05×) of HFD-fed wild-type mice. Similarly, PPARγ agonists have been shown to redistribute fat by stimulating lipid uptake and esterification potential in inguinal fat but not in perigonadal fat [25]. Thus, in the wild type, inguinal fat is the predominant depot in response to PPARγ activation, in which local insulin sensitivity is increased and ultimately contributes to whole-body metabolic homeostasis. However, in mice with partial lipodystrophy, both residual perigonadal and preserved inguinal fat depots respond to PPARγ activation, leading to a compensatory increase in the workload due to inguinal fat.

Clinically, FPLD3 patients reveal a striking paucity of subcutaneous limb and buttock fat, accompanied with “reactive” fat deposition in the viscera [3,30], which is opposite to the adipose distribution in *Pparg^C/-^* mice. Nevertheless, patients with FPLD3 and other types of familial lipodystrophy exhibit metabolic features of *Pparg^C/-^* mice, such as insulin resistance, decreased lipolysis, and inability of adipose tissue to increase triglyceride uptake after meals [3,30,31]. Intriguingly, striking lipodystrophic appearance and metabolic complications are subtle until adolescence, albeit with high fasting insulin levels in childhood [30]. Given that the metabolic homeostasis in *Pparg^C/-^* mice primarily depends on the work of inguinal fat, it is plausible to assume that the “normal” appearance of lipodystrophy patients in childhood is because the gluteal subcutaneous fat compensates for the incompetency of visceral fat. During puberty, the subcutaneous fat under chronic burden is finally exhausted and atrophies, and consequently, metabolic dysregulation emerges. Taking this pathophysiological assumption together with our finding that TZD can relieve the workload of inguinal fat, a potential therapeutic strategy for patients with familial lipodystrophy is to start TZD treatment early in childhood. Even in adult patients with mature lipodystrophy, studies have observed an improvement in metabolic factors after TZD treatment, though the accumulation of fat in lipoatrophic area was modest [32,33]. Our study suggests the benefits of TZD even in *Pparg^C/-^* mice, which are thought to have a poor response to TZD. Thus, the clinical benefits of TZD in patients with lipodystrophy might be underestimated and warrant further investigation.

There are limitations in our study. Since the gene targeting strategy in our study would affect whole-body PPARγ expression levels, it is likely that decreased PPARγ expression in beta cells would affect beta-cell function. We found that the pancreas weight relative to body weight was not distinguishable between genotypes (Appendix A). However, the contribution of beta-cell dysfunction to the overall insulin homeostasis cannot be excluded. Moreover, insulin resistance reflected by Akt phosphorylation in our study was conducted within 5 min after exogenous insulin infusion through the vena cava. While the insulin sensitivity does reflect the acute action of exogenous insulin in the current protocol, more ex vivo insulin stimulation assays from isolated adipose tissue or primary adipocytes would help address the adipose cell-autonomous role of PPARγ in insulin resistance. Another limitation of our study is that some data interpretations are based on gene changes. By using a transcriptome analysis, we can access various comparisons between depots, genotypes, and treatments. This provides us an opportunity to explore the complication of compensatory effects. Although several functional analyses have been demonstrated in our study, some conclusions are still resulted from the gene expression analysis. In the future, more functional analysis will be required to support the conclusion that inguinal depot activity is preserved in *Pparg^C/-^* mice.

By calculating the inguinal fat-dependence index, that is, the ratio of insulin sensitivity of inguinal fat to perigonadal fat, we clearly defined the relationship between perigonadal and inguinal fat. We found that TZD treatment induced the inguinal fat to overcome the increase in nutrient load, reflected by a 2.04-fold increase in the inguinal fat-dependence index in wild-type mice. In mice with partial lipodystrophy, the inguinal fat-dependence index abnormally increased to 4.55-fold owing to the decrease in insulin sensitivity of perigonadal fat, reflecting the increased workload of inguinal fat. Importantly, TZD treatment restored the insulin sensitivity of perigonadal fat, leading to a normal increase (2.78-fold) in the inguinal fat-dependence index of mice with partial lipodystrophy. This index provides a new concept for evaluating adipose tissue homeostasis, in which the workload of individual fat depots can be expressed. In summary, our results established the importance of PPARγ in controlling workloads between fat depots and contributing to adipose tissue homeostasis.

## 4. Materials and Methods

### 4.1. Animals

We generated and described the *Pparg^C/-^* mice in our previous study [13,14]. Briefly, the mice are heterozygous for the *Pparg*-C allele that carries an insertion of a c-*fos* AU-rich element (ARE) into its 3′-untranslated sequence [34]. While the transcripts of *Pparg*-C encode a normal sequence for translation, the stability of these transcripts was largely decreased in comparison with wild type allele. The *Pparg^C/-^* mice and wild-type mice were F1 littermates from mating of *Pparg^C/+^* mice on a C57BL/6J background with *Pparg^+/-^* mice on a 129S6 background (kindly provided by Dr. Ronald Evans) [7]. Male mice at 8–10 weeks of age were fed a regular chow (Purina Laboratory Rodent Diet 5001, PMI Nutrition International, Richmond, IN, USA) or a high-fat diet (58% fat energy; 58R2, TestDiet, Richmond, IN, USA). For TZD treatment, the mice were fed with a high-fat diet (58R2) blend with rosiglitazone (10 mg/kg of body weight (BW)/day). Mouse terminal surgeries were performed after intraperitoneal injection of avertin (250 mg/kg). Proper analgesia was evaluated by palpebral reflex and toe pinch reflex. The approval of the protocols was granted by the Institutional Animal Care and Use Committees of National Cheng Kung University.

### 4.2. Insulin Infusion

For the detection of insulin-induced Akt phosphorylation, 62.5 mU/kg insulin (Eli Lilly, Indianapolis, IN, USA) was administered through the vena cava, and liver, perigonadal fat, inguinal fat, and skeletal muscle were harvested at 5, 8, 10, and 15 min after insulin infusion. The harvested tissues were frozen immediately by liquid nitrogen and stored at −80 °C.

### 4.3. Immunoblot Analysis

Total protein (20 µg) form tissues were separated by SDS-PAGE, transferred to polyvinylidenedifluoride (PVDF) membranes (Pall Gelman Laboratory, Ann Arbor, MI, USA), and probed with antibodies. Immunoreactive proteins were detected using an enhanced chemiluminescence Western blotting detection system (GE Healthcare, Little Chalfont, UK). Densitometric analysis was performed using Vision WorksLS software version 6.8 (UVP, Upland, CA, USA).

### 4.4. RNA Analysis

Tissues were stored in RNAlater (Ambion, Austin, TX, USA), and total RNA was extracted with REzol (Protech Technology, Taipei, Taiwan). Samples of mRNA were analyzed with SYBR Green-based quantitative real-time polymerase chain reaction (qRT-PCR), with β-actin as the reference gene in each reaction.

### 4.5. Glucose Tolerance Test

Mice were fasted for 4–6 h and given glucose (2 g/kg of BW) by oral gavage. Blood samples were collected before and at the times indicated after injections. Plasma glucose concentration was determined by a glucose colorimetric test (Autokit Glucose, Wako, Richmond, VA, USA). Insulin was measured using a mouse insulin ELISA (Ultrasensitive Mouse Insulin ELISA, Mercodia AB, Uppsala, Sweden). The insulin-resistance (IR) index was calculated as the product of the areas under glucose and insulin curves in glucose tolerance tests as previously described [13].

### 4.6. Explant Culture of Adipose Tissue and Lipolysis

Adipose tissues were collected from mice at 8 weeks of age, washed by PBS, cut into pieces 0.2–0.3 cm^3^ in size and cultured in DMEM. The next day, tissues were treated with 50 μM isoproterenol or vehicle for 4 h and collected the medium to analyze for free fatty acid concentration. Serum free fatty acid was measured by a colorimetric kit (NEFA C; Wako, Richmond, VA, USA).

### 4.7. Surgical Removal of Inguinal Fat

Isoflurane anaesthetized mice were placed on a heating pad in a supine position. The hair was removed from the incision area, and the area was sterilized with a 70% ethanol and povidone-iodine. Short (1 cm) skin incisions were made bilaterally, beginning at the ventral side close to the proximal end of the hind limb and continuing parallelly to vertical line of body. The inguinal fats were exposed and removed, and the silk sutures were made. The mice were rested for two days.

### 4.8. Fibrosis Staining

Adipose tissues were fixed in 4% para-formaldehyde and embedded in paraffin. Sections with a thickness of 10 μm were cut and stained with Masson’s trichrome or picrosirius red.

### 4.9. Data Analysis

Values are shown as mean ± SEM. Statistical analyses were conducted by Student’s *t* test or one-way ANOVA (Figure 4) followed by Fisher’s least significant difference test. Student’s *t* test was used for comparisons between *Pparg*^C/-^ mice and wild-type mice. Differences were considered to be statistically significant at *p* < 0.05.

## Figures and Tables

**Figure 1 ijms-24-03904-f001:**
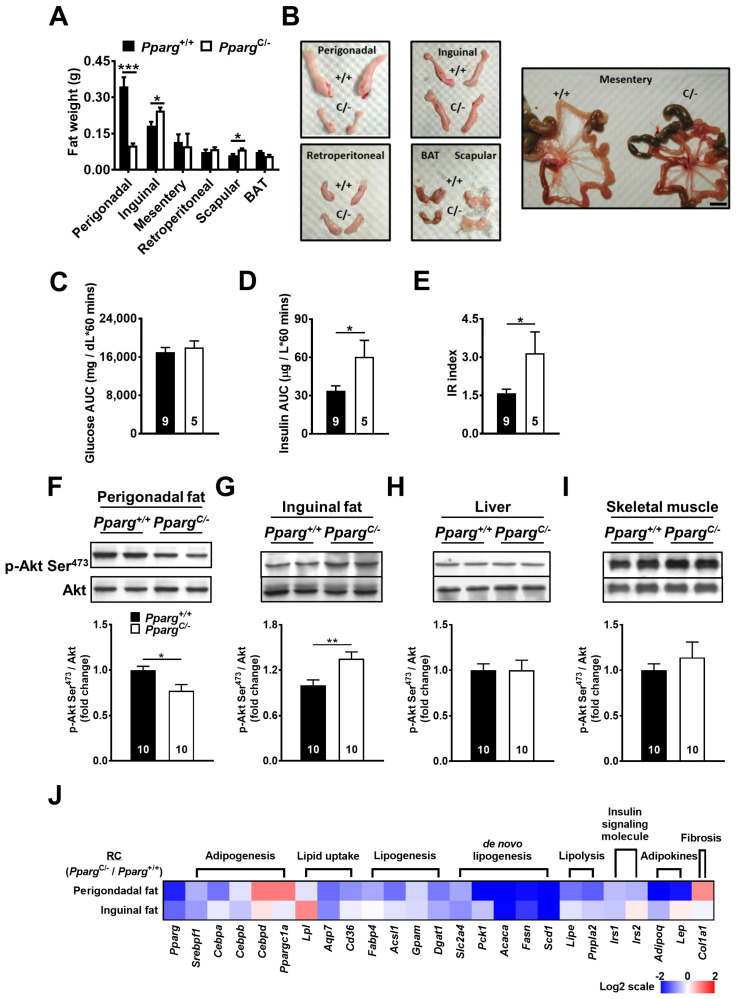
Insulin sensitivity and mRNA levels of metabolic genes in *Pparg^C/-^* mice. (**A**) The fat weight of various depots in *Pparg*^+/+^ and *Pparg*^C/-^ mice. (**B**) Gross appearance of individual fat depots of 4-month-old *Pparg^C/-^* mice and *Pparg^+/+^* mice. Scale bar = 1 cm. Plasma glucose (**C**) and insulin (**D**) area under curve (AUC) and insulin resistance (IR) (**E**) index during GTT in *Pparg*^+/+^ and *Pparg*^C/-^ mice. Immunoblot analyses of Akt phosphorylation in perigonadal fat (**F**), inguinal fat (**G**), liver (**H**), and skeletal muscle (**I**) of *Pparg*^+/+^ and *Pparg^C/-^* mice after insulin stimulation. The intensities of the bands are quantified densitometrically relative to *Pparg*^+/+^ mice. Data are expressed as means ± SEM, *n* = 10 in each group. * *p* < 0.05, ** *p* < 0.01, and *** *p* < 0.001 compared with *Pparg*^+/+^. (**J**) The heat-map represents mRNA levels of metabolic genes analyzed by quantitative real-time PCR in perigonadal and inguinal fat of *Pparg^C/-^* mice relative to *Pparg*^+/+^ mice. The log 2 expression scales represent the values in terms of logarithmic fold, blue (lowest), white (medium), and red (highest) expression levels. RC, regular chow.

**Figure 2 ijms-24-03904-f002:**
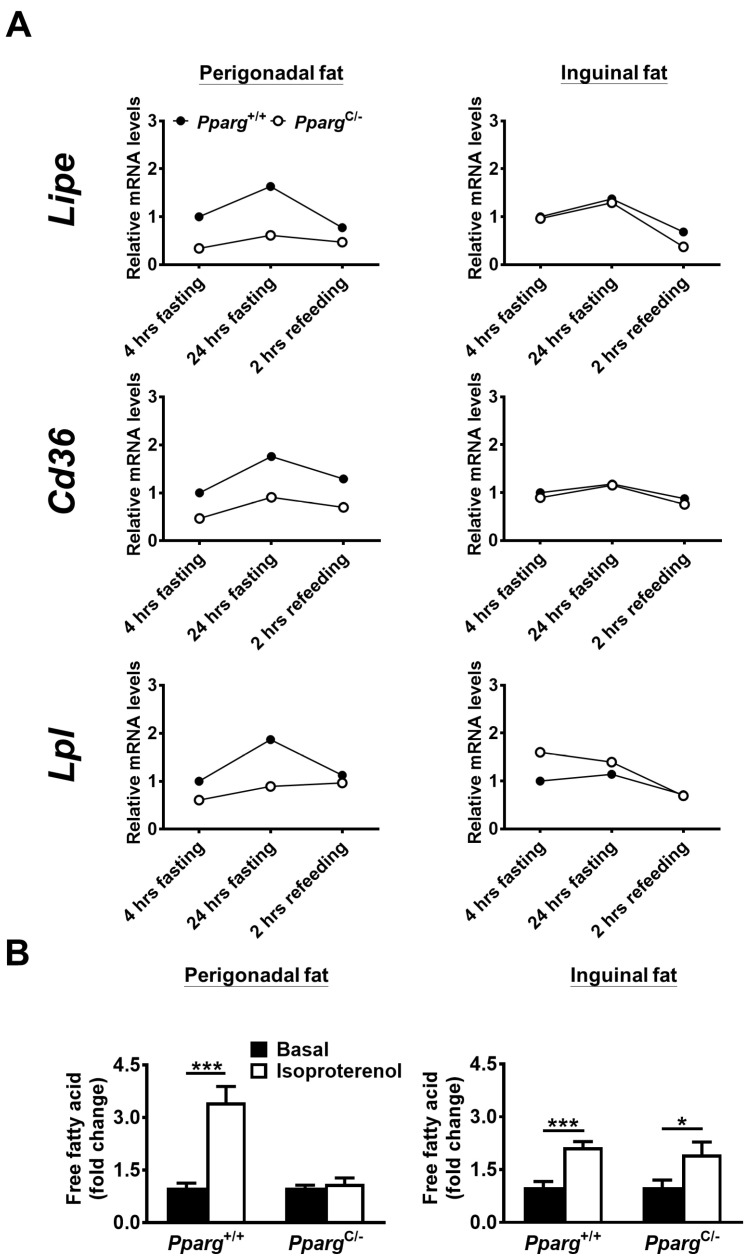
Adipose tissue flexibility and expression of genes related to lipid mobilization of fat tissues in *Pparg^C/-^* mice. (**A**) mRNA levels of genes related to lipid mobilization in perigonadal (left panels) and inguinal (right panels) fat of chow-fed *Pparg^C/-^* and *Pparg*^+/+^ mice after 4 h fasting, 24 h fasting, or 2 h refeeding. The mRNA levels of *Pparg*^+/+^ after 4 h fasting were used as the relative reference. Data are expressed as means. (**B**) The secretion levels of free fatty acids of fat depots isolated from *Pparg^C/-^* mice and *Pparg*^+/+^ mice in the presence or absence of isoproterenol stimulation. Data are expressed as means ± SEM. * *p* < 0.05 and *** *p* < 0.001 compared between treatment.

**Figure 3 ijms-24-03904-f003:**
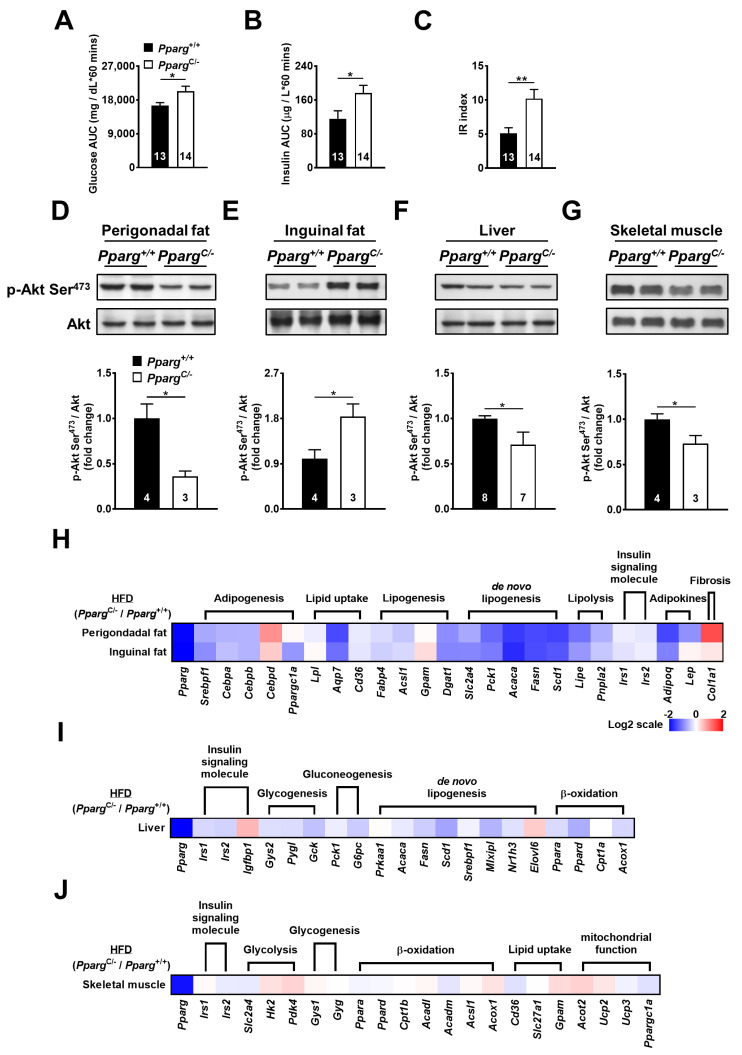
Insulin sensitivity and mRNA levels of metabolic genes in HFD-fed *Pparg^C/-^* mice. Plasma glucose (**A**) and insulin (**B**) area under curve (AUC) and insulin resistance (IR) (**C**) index during GTT in HFD-fed *Pparg*^+/+^ and *Pparg*^C/-^ mice. Immunoblot analyses of Akt phosphorylation in perigonadal fat (**D**), inguinal fat (**E**), liver (**F**), and skeletal muscle (**G**) of *Pparg*^+/+^ and *Pparg^C/-^* mice after insulin stimulation. The intensities of the bands are quantified densitometrically relative to *Pparg*^+/+^ mice. Data are expressed as means ± SEM, *n* = 3–8 in each group. * *p* < 0.05 and ** *p* < 0.01 compared with *Pparg*^+/+^ mice. (**H**-**J**) The heat-map represents mRNA levels of metabolic genes analyzed by quantitative real-time PCR in perigonadal fat, inguinal fat, liver, and skeletal muscle tissues of *Pparg^C/-^* mice relative to *Pparg*^+/+^ mice. The log 2 expression scales represent the values in terms of logarithmic fold, blue (lowest), white (medium), and red (highest) expression levels. HFD, high-fat diet.

**Figure 4 ijms-24-03904-f004:**
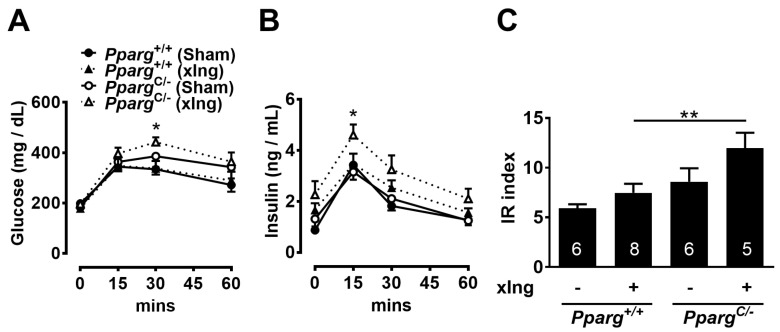
Effects of inguinal fat removal on insulin sensitivity in HFD-fed *Pparg^C/-^* mice. Plasma glucose (**A**) and insulin (**B**) levels and IR index (**C**) during GTT of HFD-fed *Pparg^C/-^* mice and *Pparg*^+/+^ mice with surgical removal of inguinal fat for 3 weeks. Data are expressed as means ± SEM, *n* = 5–9 in each group. * *p* < 0.05 and ** *p* < 0.01 compared between inguinal fat-removed *Pparg^C/-^* and *Pparg*^+/+^ mice. xIng, inguinal fat-removed mice.

**Figure 5 ijms-24-03904-f005:**
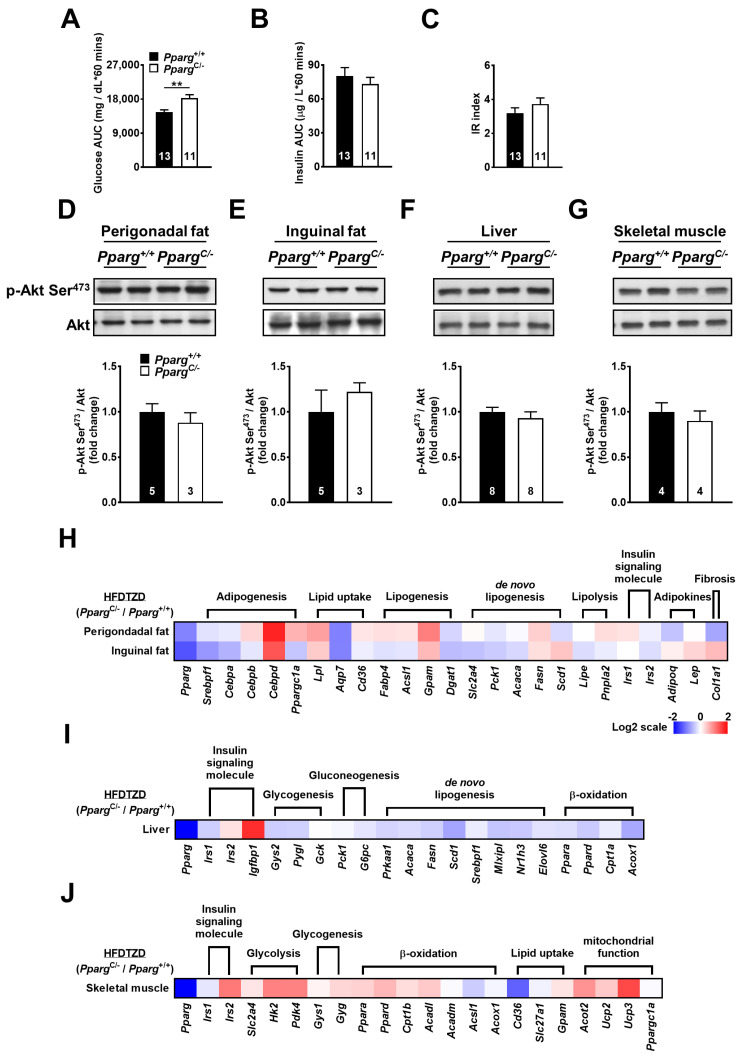
Insulin sensitivity and mRNA levels of metabolic genes in HFDTZD-fed *Pparg^C/-^* mice. Plasma glucose (**A**) and insulin (**B**) area under curve (AUC) and insulin resistance (IR) (**C**) index during GTT in HFDTZD-fed *Pparg*^+/+^ and *Pparg*^C/-^ mice. Immunoblot analyses of Akt phosphorylation in perigonadal fat (**D**), inguinal fat (**E**), liver (**F**), and skeletal muscle (**G**) of *Pparg*^+/+^ and *Pparg^C/-^* mice after insulin stimulation. The intensities of the bands are quantified densitometrically relative to *Pparg^+/+^*. Data are expressed as means ± SEM, *n* = 3–8 in each group. ** *p* < 0.01 compared with *Pparg^+/+^*. (**H**–**J**) The heat-map represents mRNA levels of metabolic genes analyzed by quantitative real-time PCR in perigonadal fat, inguinal fat, liver, and skeletal muscle tissues of *Pparg^C/-^* mice relative to *Pparg*^+/+^ mice. The log 2 expression scales represent the values in terms of logarithmic fold, blue (lowest), white (medium), and red (highest) expression levels. HFDTZD, high-fat diet supplemented with TZD.

**Figure 6 ijms-24-03904-f006:**
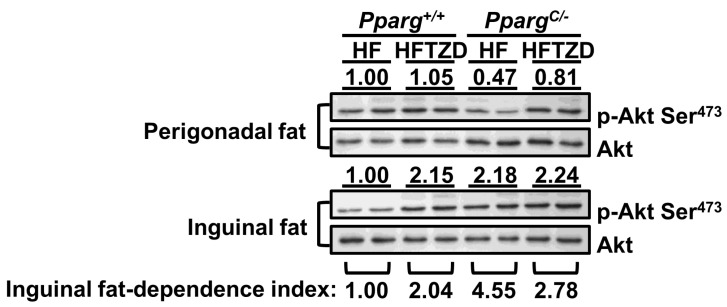
Effects of TZD on insulin sensitivity and demonstration of inguinal fat-dependence index in perigonadal and inguinal fat. Immunoblot of Akt phosphorylation is compared in fat tissues with or without TZD treatment. The Akt phosphorylation level in perigonadal or inguinal fat of HFD-fed *Pparg^+/+^* mice is used as the reference. The inguinal fat-dependence index is calculated as the relative Akt phosphorylation level of inguinal fat divided by the relative Akt phosphorylation level of perigonadal fat in *Pparg^C/-^* mice and *Pparg^+/+^* mice with or without TZD treatment.

## Data Availability

Not applicable.

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
