# Peer review of "Inguinal Fat Compensates Whole Body Metabolic Functionality in Partially Lipodystrophic Mice with Reduced PPARγ Expression"

_ijms, 2023, doi:10.3390/ijms24043904_

Round 1

Reviewer 1 Report

Review of “Inguinal fat compensates whole body metabolic functionality in partially lipodystrophic mice with reduced PPARγ expression” by Cherng-Shyang Chang et al.

This is an interesting study of PPARγ mutant mice. The authors demonstrated that inguinal fat of PpargC/- mice compensate abnormalities of perigonadal fat. Overall, it is well written and the data seems supports the conclusions. Since PPARγ mutations in humans are relevant to insulin resistance and lipodystrophy, these findings may have a clinical impact. Below are some suggestions for improve.

1.     Insulin resistance is a major phenotype of this research. The authors showed clear immunoblot of p-Akt which nicely supporting the conclusions. In addition to this, expressions of insulin/Akt downstream genes should be upregulated or downregulated depends on whether insulin promotes or inhibits them. Irs1 and Irs2 are regulated by phosphorylated rather than transcription thus would not be optimal to assess the activity of insulin signaling. Besides, Igfbp1 is not a good candidate either since it is binding to insulin and insulin like growth factors, act as an upstream regulator. These qPCRs should be replaced by data of insulin/Akt downstream genes.

2.     The supplemental figure of fat tissues should be in main figures. They showed perigonadal fat was affected by the mutation more than other fat tissues. This might be the reason why inguinal fat could compensate. One question regarding this is that other other visceral depots, such as mesentery and retroperitoneal fat in PpargC/- mice were not altered. Do they play a compensatory role in combating perigonadal fat abnormalities as well? This needs to be documented or discussed.

Minor comments

1.     qPCR results are all showed in the ratio of mutant/wildtype which eliminated lots of information. Figure showing individual levels of each gene in each condition compared with wildtype should be provided, e.g. in supplemental figure.

2.     Figure 3F, an error bar is missing.

3.     PPAR agonist TZD, the full name of TZD should be include at the first time using TZD.

4.     Line 165-166, “The genes encoding insulin signaling (IRS1 and IRS2) and adipokines (adiponectin and leptin) in adipose tissue were…” should be “IRS1 and IRS2 encoding two insulin receptor substrate proteins”. There is nothing could encode insulin signaling, nor lipogenesis (line 155-156), lipolysis (163), adipogenesis (214).

Reviewer 2 Report

In this manuscript, Chang et al address the role of fat depots in partial lipodystrophy in metabolic homeostasis. The manuscript is well done; however, it could be improved by further experiments to support the conclusion.   

Line 119-120: The authors state that “ plasma insulin levels of PpargC/- mice were increased, leading to a significant increase in insulin resistance” however, multiple reports a role of ppary in beta cells.  The authors should consider how the decreased expression of PPARy affects beta cell function. Increased insulin may contribute to insulin resistance but an adipose cell-autonomous role of pparg in insulin resistance maybe the driver of the phenotype.

Figure 1I: data interpretation needs more support to conclude that inguinal depot activity is preserved for example:

Line 159-160: “Of particular importance is the significant increase in the expression of LPL, which suggests a significant increase in lipid uptake” this is based on one gene change.

Col1a1 expression alone is not sufficient to claim increased fibrosis.

Histological assessment of the tissues in both LFD and HFD conditions will strengthen the conclusion of Figures 2 and 3.

Figure 2 examines lipolysis. Please change the title.

Figure 4:  what are the effects under chow diet? are there any changes in insulin sensitivity or plasma insulin levels?  
